# Labor curves based on cervical dilatation over time and their accuracy and effectiveness: A systematic scoping review

Johanne Mamohau Egenberg Huurnink[1,2]*, Ellen Blix[2], Elisabeth Hals[3], Anne Kaasen[2], Stine Bernitz[2,4], Tina Lavender[5], Mia Ahlberg[6], Pål Øian[7], Aase Irene Høifødt[1], Andrea Solnes Miltenburg[8], Aase Serine Devold Pay[1,2,9]

1 Department of Obstetrics and Gynecology, Oslo University Hospital, Oslo, Norway, 2 Faculty of Health Sciences, Oslo Metropolitan University, Oslo, Norway, 3 Department of Obstetrics and Gynecology, Innlandet Hospital Trust, Lillehammer, Norway, 4 Department of Obstetrics and Gynecology, Østfold Hospital Trust, Grålum, Norway, 5 Department of International Public Health, Centre for Childbirth, Women's and Newborn Health, Liverpool School of Tropical Medicine, Liverpool, United Kingdom, 6 Department of Medicine, Clinical Epidemiology Division, Solna, Karolinska Institutet, Stockholm, Sweden, 7 Department of Obstetrics and Gynecology, University Hospital of North Norway, Tromsø, Norway, 8 Department of Obstetrics and Gynecology, Akershus University Hospital, Akershus, Norway, 9 Department of Obstetrics and Gynecology, Bærum Hospital, Vestre Viken Hospital Trust, Bærum, Norway

* jme.huu@gmail.com

**Data Availability Statement:** All relevant data are within the manuscript and its Supporting Information files.

## Abstract

### Objectives

This systematic scoping review was conducted to 1) identify and describe labor curves that illustrate cervical dilatation over time; 2) map any evidence for, as well as outcomes used to evaluate the accuracy and effectiveness of the curves; and 3) identify areas in research that require further investigation.

### Methods

A three-step systematic literature search was conducted for publications up to May 2023. We searched the Medline, Maternity & Infant Care, Embase, Cochrane Library, Epistemonikos, CINAHL, Scopus, and African Index Medicus databases for studies describing labor curves, assessing their effectiveness in improving birth outcomes, or assessing their accuracy as screening or diagnostic tools. Original research articles and systematic reviews were included. We excluded studies investigating adverse birth outcomes retrospectively, and those investigating the effect of analgesia-related interventions on labor progression. Study eligibility was assessed, and data were extracted from included studies using a piloted charting form. The findings are presented according to descriptive summaries created for the included studies.

### Results and implications for research

Of 26,073 potentially eligible studies, 108 studies were included. Seventy-three studies described labor curves, of which ten of the thirteen largest were based mainly on the United States Consortium on Safe Labor cohort. Labor curve endpoints were 10 cm cervical

**Funding:** The author(s) received no specific funding for this work.

**Competing interests:** The authors have declared that no competing interests exist.

dilatation in 69 studies and vaginal birth in 4 studies. Labor curve accuracy was assessed in 26 studies, of which all 15 published after 1986 were from low- and middle–income countries. Recent studies of labor curve accuracy in high-income countries are lacking. The effectiveness of labor curves was assessed in 13 studies, which failed to prove the superiority of any curve. Patient-reported health and well-being is an underrepresented outcome in evaluations of labor curves. The usefulness of labor curves is still a matter of debate, as studies have failed to prove their accuracy or effectiveness.

## Introduction

Intrapartum care is an art of balance in which physiologic birth is supported and medical interventions accelerating progression should be available when "indicated." The established indicators for intervention due to prolonged labor are labor curves, which graphically visualize cervical dilatation over time; the most commonly used are the alert line (1 cm/hour) and action line (parallel to the alert line) in the World Health Organization's (WHO's) partograph [1]. Despite global implementation, research in recent decades has failed to prove the ability of these indicators to improve birth outcomes and accurately identify those at risk of adverse birth outcomes [2,3]. Oladapo et al. [4] concluded that the alert line's progression rate of 1 cm/hour was unrealistically rapid for most nulliparous and multiparous women. Despite the paucity of evidence, however, these labor curves remain central decision-making tools in obstetric and midwifery practice for low- and high-risk patients worldwide, and no superior tool has been verified. When the established indicators for interventions in cases of prolonged labor prove ineffective in enhancing birth outcomes, it disrupts the delicate balance of intrapartum care. This imbalance can lead to an increased risk of both over- and underuse of medical interventions, jeopardizing the overall quality of labor care.

Labor progression has been visualized with curves since the 1950s [5,6]. Friedman [5,6] derived a sigmoid labor curve that was adopted and adjusted by Philpott and Castle [7,8]. Philpott & Castle created the partograph featuring linear labor curves for 1 cm/hour progression, with the aim of providing a tool that could be used feasibly to distinguish women with "abnormal" labor from the majority of "normal laboring women" [7,8]. The WHO adopted the partograph and conducted a large multicenter study to examine its effectiveness in the 1990s [1]. In addition to labor curves, the partograph has components prompting the measurement and recording of maternal characteristics and vital signs, contractions, fetal heart rate, and treatment provided. The study yielded promising results suggesting that partograph use improved labor outcomes [1], and the partograph was promoted and implemented globally as the gold standard for labor monitoring [9–12]. The study, however, has been criticized due to its sole focus on the partograph as the intervention, when it also included healthcare personnel training in what could be interpreted as a complex intervention [13]. In the last two decades, new attempts have been made to understand the patterns of labor. Central contributors to the development of contemporary labor curves include Zhang et al. [14], who in 2010 presented a curve based on data from a large United States (US) cohort and the application of new statistical approaches. This curve accelerates from 5–6 cm cervical dilatation, thereby providing a new threshold for the onset of active labor, and features stepped lines based on the time taken to progress from one integer centimeter to the next [14].

Labor curves are implemented in diverse contexts worldwide, and research thereon faces challenges related to the heterogeneity of curve designs, guidelines, contexts, methodologies,

and levels of care [15,16]. Labor progression is extremely variable, and "one-size-fits-all" curves fail to distinguish those at risk of adverse birth outcomes [3]. In 2018, the WHO published recommendations for "intrapartum care for a positive childbirth experience" [16] that include a revision of the partograph as a priority. In 2020, the WHO introduced the Labor Care Guide (LCG), a monitoring-to-action tool derived from an emerging body of evidence that includes a guideline for expected labor progression according to the time taken to move from one integer centimeter of cervical dilatation to the next, allowing for individual variation in labor progression and the adoption of a more woman-centered approach [17,18]. Large studies exploring the effect of LCG use on birth outcomes are planned, but no publication has appeared to date [19–21].

The scoping review approach enables extensive mapping of the evidence of interest, regardless of study heterogeneity [22]; it allows for the inclusion of studies with different designs conducted in different contexts, and complements data from the most-cited studies and systematic reviews. Labor curves play persistent roles in obstetric and midwifery care globally, and the search for optimal tools for labor progression monitoring provides the rationale for the performance of this review.

This systematic scoping review aimed to give an overview of studies providing labor curves for describing cervical dilatation over time. The specific aims were to 1) identify and describe labor curves; 2) map any evidence for, as well as outcomes used to evaluate the accuracy and effectiveness of the curves; and 3) identify areas in research that require further investigation.

## Methods

The study protocol (S5 File), based on the Joanna Briggs Institute's (JBI's) framework for scoping review development [22], was registered in Open Science Framework https://osf.io/2nzb3/ (protocol ID: 2NZB3) and published prior to literature search initiation [23]. This study is reported according to the Preferred Reporting Items for Systematic Reviews and Meta-Analyses extension for scoping reviews (S4 File) [24].

In July 2020, a systematic three-step literature search was initiated, and most publications were retrieved within January 2021. First, a senior librarian searched the Medline and Maternity & Infant Care databases broadly to identify relevant search terms (S1 File). Second, the librarian searched the Medline, Maternity & Infant Care, Embase, Cochrane Library, Epistemonikos, CINAHL, Scopus, and African Index Medicus databases using multiple combinations of search terms, due to the wide range of terminology used to describe labor progression and to ensure the sensitivity of the search (S2 File). A different methodology was used to search the African Index Medicus due to the limitations of the database (S2 File). The eligibility of retrieved publications was assessed, and the full texts of included studies were retrieved. In the third step, two researchers (ASDP and JMEH) searched the reference lists of these studies to identify additional sources of evidence. In May 2023, a comprehensive update of the literature search was conducted. Source selection was managed using the Covidence systematic review software (Veritas Health Innovation Ltd., Melbourne, Australia), a web-based collaboration software platform.

Only published research articles and systematic reviews were eligible for inclusion. We used the mnemonic PCC (population, concept, and context) to define the review's focus and eligibility criteria [22]. The eligible *population* comprised nulli- and multiparous women with spontaneous and induced labor and infants with and without risk factors and complications. The *concept* evaluated was labor curves describing cervical dilatation against time during active labor. Studies describing and/or assessing the accuracy and/or effectiveness of these curves were eligible. Accuracy was defined as the ability to identify those at risk of adverse birth

outcomes (i.e., screening) or prolonged labor requiring intervention (i.e., diagnosis). Effectiveness was defined as the potential to improve birth outcomes (e.g., maternal and/or neonatal morbidity or mortality, need for obstetric intervention, or patient-reported satisfaction, health, and/or well-being). The *contexts* were healthcare settings in which skilled birth attendants provided care to women in labor.

We excluded studies designed specifically to investigate adverse birth outcomes (e.g., retrospective studies where all participants had adverse birth outcomes such as uterine rupture or severe perineal lacerations prior to inclusion) and those investigating the effect of analgesia-related interventions on labor progression. No limitation on the date of publication was imposed. In accordance with the JBI's recommendation [22], the quality of the included articles was not assessed [23].

Seven reviewers (AIH, AK, ASDP, ASM, EB, EH, and JMEH) participated in the screening of the titles and abstracts of potentially eligible publications, after pilot testing to increase the inter-reviewer consistency of source selection. Two reviewers independently screened each publication, and any disagreement on eligibility was resolved by a third reviewer. Articles in languages other than Norwegian, Swedish, Danish, English, Dutch, Icelandic, German, and Kiswahili were excluded at this point. ASDP and JMEH screened most full texts and EB and JMEH screened articles in German for eligibility, with any disagreement resolved by consensus.

The seven reviewers performed data charting after pilot testing of the preliminary data charting form and revision to create the final version (S3 File) [23]. The form was not altered further during data extraction. The following data were extracted from the included studies: country, study type and aim, population size and characteristics, methods, results, definition of the first stage of active labor, outcomes measured, and other relevant information. Each included study was reviewed by two reviewers, with one reviewer extracting the data, a second reviewer verifying it, and the final data confirmed by consensus between the reviewers.

The data charting forms served as the foundation for evidence mapping. The articles were grouped according to whether they described the labor curves or their accuracy or effectiveness. Characteristics of included studies were described in S1 Table by author, year, country, design, study population and information of interest. We created descriptive summaries of the included studies to answer the research questions. The findings were presented by regions defined in the United Nations' Sustainable Development Goals [25].

## Results

The search identified 26,073 potentially relevant citations, of which 9246 were excluded as duplicates. The titles and abstracts of the remaining 16,827 publications were screened, leading to the exclusion of 16,391 publications. The full texts of 435 articles were screened, leading to the exclusion of 327 articles, mainly because they were unrelated to the topic of interest or were not research articles (e.g., commentaries and posters). In total, 108 studies were included in the review (Fig 1): 73 described labor curves [4,7,14,26–95], 26 assessed curve accuracy [2,3,7,8,31,32,81,96–114], and 13 assessed curve effectiveness [1,15,115–125]. 4 studies described both labor curves and their accuracy [7,31,32,81].

### Descriptions of labor curves

Of the 73 studies describing labor curves, 47 were from Europe and North America [14,27,29,30,33–42,45,46,48,50–55,57–59,61,64,66,68,69,71,74,77–81,83–86,90–93,95], 8 were from East and Southeast Asia [31,49,56,67,75,76,82,94], 7 were from North Africa and Western Asia [26,43,47,63,72,73,89], 5 were from Sub-Saharan Africa [7,60,65,87,88], 4 were from

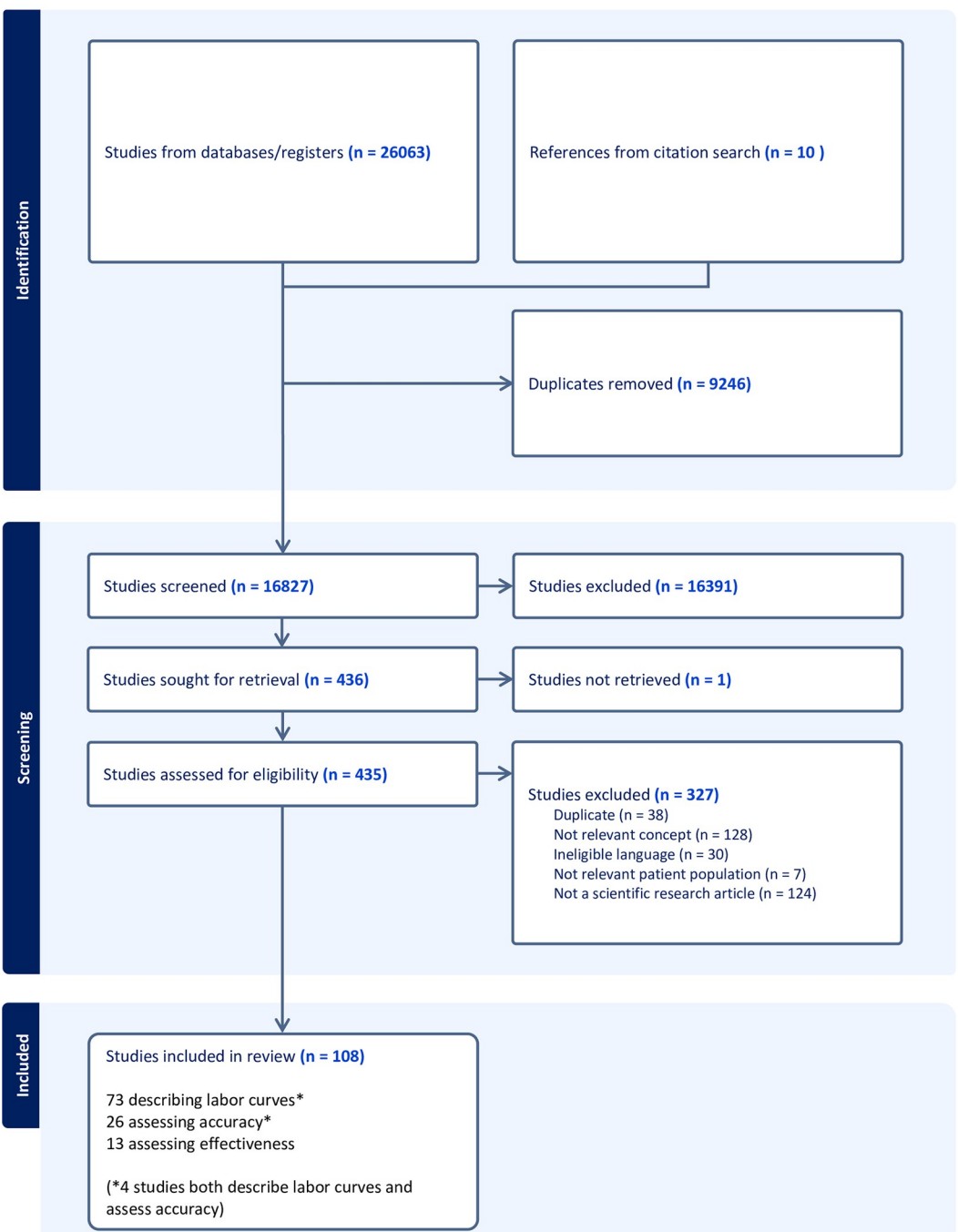

**Fig 1. PRISMA flow diagram of literature search and study selection.**

Central and South Asia [28,32,62,70], and 2 studies were conducted in multiple regions [4,44]. The date range was from 1951–2023.

Twenty of these studies were based on data from three US cohorts: a cohort in St. Louis, Missouri [29,30,42,45,61,64, 66,78,79,84], the large multicenter Consortium on Safe Labor (CSL) cohort [14,34,41,52,55,58,66,83,91], and the National Collaborative Perinatal Project (CPP) cohort [92]; the CSL and CPP cohorts were compared in one study [53]. The thirteen largest studies were conducted with 50,000–150,000 participants and comprised a systematic review and meta-analysis including CSL cohort data [4], eight studies based solely on CSL data [14,34,41,52,55,66,83,91], a study in which the CSL and CPP cohorts were compared [53], two studies based on Israeli cohort data [43,47], and a study based on Swedish cohort data [59]. Most studies had smaller samples: 30 studies included <1000 women [7,27,31,32,35–39,46,51,56,57,60,63,67–70,72–74,77,78,80,81,86,89,90,94], 26 studies included 1000–10,000 women [28,30,33,42,44,45,48–50,54,58,61,62,64,65,71,75,76,79,82,84,85,87,88,93,95], and 4 studies included 10,000–50,000 women [26,29,40,92].

Most of these studies were conducted with women with singleton pregnancies and fetuses in cephalic presentation who delivered at term or delivered infants weighing > 2.5 kg after reaching 10 cm cervical dilatation (S1 Table). Labor curve comparison was challenging due to the heterogeneity in definitions of antenatal risk factors, intrapartum risk factors and treatments, and adverse outcomes (S1 Table). The curves were stratified by parity [4,14,26–30,33–35,38,43–47,49–60,62,64,65,67,69,71,72,74,76,80,81,83–85,89–92,95], labor onset [29,41,45,47,48,57,58,78], body mass index [29,52,64,74,83,86], maternal age [61,91,95], ethnicity [33,60,84], fetal sex [30,62], centimeters of cervical dilatation on admission [14,75,81], fetal size [29,55,83], epidural analgesia use [26,47,63], fetal presentation [27,38,40]. In some studies, study group inclusion and stratification were based on specific characteristics: varieties of trial of labor after caesarean section (CS) [41,42,63,73, 94], preterm induction of labor [34], cephalopelvic disproportion [39], multiple gestations [72,77], cervical cerclage [66,89], and diabetes status [29,83]. Three studies included preterm gestations down to 34 weeks [47,50,55], and three included gestations down to 23–24 weeks [77–79], of which one with the derived curves stratified by gestational age [79].

Labor progression was presented in these studies as graphs of cervical dilatation against time ([7,14,26–47,49–84,87–95], as the time taken to move from one integer to the next [4,14,26–28,42,43,45,47–49,52,55,58,59,61,66,70,76,78,79,82–86,91–94], and as cumulative stepped lines based on 95th percentile traverse times with starting point cervical dilatation on admission [14,26,59]. Few labor curves defined normality or abnormality; most were based on the central tendencies mean or median [7,14,26–47,49–84,87–95], and some also presented other percentiles [35,50,59,60]. Labor curve endpoints were the first cervical dilatation measure of 10 cm (representing the end of first stage of active labor) in 69 studies [4,7,14,26–66,70,72–95] and the time of birth (representing the first and second stages of active labor) in four studies [67–69,71]. The initiation of the first stage of active labor was defined by different parameters, often including various limits of cervical effacement, contractions, and cervical dilatation. The cervical dilatation on first vaginal examination, rather than a specific value, served as the starting point of some labor curves [28,31,33,46,63,73,75,81].

## Accuracy of labor curves in predicting birth outcomes

Of the 26 studies evaluating the accuracy of labor curves, 11 were from Central and South Asia [32,97,102–104,106–110,113], 7 were from Sub-Saharan Africa [3,7,8,99,100,105,114], 5 were from Europe and North America [81,98,101,111,112], one was from Oceania [96], one was from East and Southeast Asia [31], and one was a systematic review including studies from

multiple countries [2]. The studies were published between 1972 and 2022. The most recent study from a high-income country was conducted in Taiwan and published in 1986 [31]. Most study populations comprised <1000 women [7,8,31,32,81,97,98,101–111,113,114]; four comprised 1000–3500 women [96,99,100,112], one comprised 9995 women [3], and that of the systematic review comprised 20,471 women [2].

The accuracy of the alert and action lines in the WHO partograph was assessed in several studies [2,3,97,100,102–110,113,114], some of which were included in the systematic review conducted for that purpose by Bonet et al. [1–3,97,100,105–108,110, 114]. In addition, the accuracy of Studds' labor stencil [81,98,111,112], Philpott and Castle's partograph [7,8,96,99], Hendricks' labor curve [101], and two curves originally described in the same articles [31,32] was assessed. The accuracy of the WHO partograph alert line and subsequent lag times for women who had previously undergone CS was assessed in two studies [102,103]. In one study, the accuracy of the WHO partograph alert and action lines was assessed for a low-risk population divided according to where the "cervix dilatation and descent curve" fell relative to these lines [104]. We understand this to mean merely the cervical dilatation curve, as the descent curve cannot be compared with the lines representing expected cervical dilatation. No study assessing the accuracy of Zhang et al.'s [14] labor curve was identified.

Perinatal and maternal outcome measures were used to assess the accuracy of labor curves. The most common were the mode of birth [7,31,32,81,96,98,101,102,104–112,114] and Apgar score (by means or <5–7 at 1 or 5 minutes) [2,8,81,98,99,101,106,111,112,114]. Other common perinatal outcomes measured were resuscitation after birth [2,97,99,100,114], admission to the neonatal intensive care unit [98,104,113], perinatal death (defined as "fresh stillbirth," "labor ward death," and "macerated stillbirth") [2,99,100,105,114], and birth asphyxia [2,105,106,108]. Other maternal outcomes measured were the need for interventions such as oxytocin augmentation [32,97,101,103,105,108,112] and epidural analgesia [101,111,112]. Maternal morbidity was measured mainly as uterine rupture or scar dehiscence [2,102,103] and organ dysfunction with dystocia [2,3]. Maternal mortality was reported in two of the included studies [2,99].

In their systematic review, Bonet et al. [2] reported that a wide range (8–76%) of women cross the WHO's alert line, regardless of maternal and perinatal outcomes. They found that no study had yielded a robust diagnostic accuracy profile for any selected outcome [2]. They reported sensitivities and specificities for the 1 cm/hour alert line of 36.0–100.0% and 24.1–91.1%, respectively, for the prediction of stillbirth during labor and 50.0–100.0% and 22.8–87.2%, respectively, for the prediction of Apgar score < 7 at 5 minutes.

Sixteen of the studies assessing the accuracy of labor curves [7,8,31,32,81,96,98,99,101–104,109–113] were not included in Bonet et al.´s [2] review. These studies assessed several different labor curves. Those that assessed the WHO's alert and action lines concluded that the lines have predictive value for the detection of abnormal labor requiring management [104,109,113], and (with minor modification) probably also to decrease the risk of uterine rupture for those who had previously undergone CS [102,103]. The studies of Studd's labor curve concluded that the nomogram [81,111,112] and labor patterns [98] can predict abnormal labor and ensure the judicious use of oxytocin. This labor curve was found to indicate the correct time for referral and aid the recognition of high-risk labor [111], and the use of the action line was found to abolish prolonged labor as a cause of stillbirth [112]. Philpott and Castle's labor curves were determined to efficiently distinguish women who can safely deliver vaginally and those at risk of severe morbidity and mortality [7,8,96,99]. Chen and Chu [31] and Daftary and Mhatre [32] concluded that their respective labor curves were efficient screening tools for the early identification of labor dystocia. The sample size in Hunters study was too small to assess the accuracy of Hendricks labor curve in predicting low Apgar score [101].

## Effectiveness of labor curves on birth outcomes

Of the 13 studies evaluating the effectiveness of labor curves, 4 were from Norway [115–117,124]; 2 each were from England [118,119], Australia [120,121], and India [123,125]; 1 each was from Southeast Asia (Indonesia, Malaysia, and Thailand) [1] and Nigeria [122]; and 1 was a systematic review including studies from multiple countries [15]. The studies were published in 1994–2023, with the earliest being the WHO's longitudinal randomized controlled trial (RCT) comparing partograph designs [1]. This study was followed by two RCTs conducted in England and published in 1998 [118] and 2006 [119], and 10 studies published between 2017 and 2023 [15,115–117,120–125]. Six studies each had <1000 participants [118,120–123,125] and 1000–10,000 participants [15,115–117,119,124], and one study had 35,484 participants [1].

Most outcomes used to assess the effectiveness of labor curves were related to obstetric interventions, such as the mode of birth [1,15,115,118–123,125], oxytocin augmentation [1,15,115,116,118,119,121–123,125], and artificial rupture of membranes (ARM) [15,115,118,119,121,125]. Maternal outcomes included the duration of labor [1, 15,117–119,121–123], postpartum hemorrhage or blood transfusion as a proxy [15,115,118,119,121–123], maternal satisfaction or childbirth experience [15,118,119,121,122,124], and perineal trauma [15,115,120]. Perinatal outcomes included the Apgar score [15,115,118–122,125] and umbilical cord pH [15,115,118–121], in some studies as part of composite neonatal outcomes [120,121].

In a Cochrane review of 11 studies, partograph designs were assessed against each other and against no partograph use [15]. The WHO partograph was most commonly tested, but no design proved to be superior to another and the evidence was insufficient to establish the effectiveness of partograph use in terms of birth outcomes [15]. Four studies included in this scoping review [118,119,121,125] were included in the Cochrane review [15]. In three of them, the effectiveness of WHO action line placement 2, 3, and 4 hours to the right of the alert line was assessed [118,119,125]. In the fourth, the WHO 4-hour action line was compared with Neal and Lowe's stepped dystocia line [121].

Of the eight studies not included in the Cochrane review, one was the 1994 WHO trial [1] and the others were published after the review [115–117,120,122–124]. In the WHO study, partograph effectiveness was assessed by comparing pre–and post–partograph implementation outcomes in a Southeast Asian population; fewer labors of >18 hours, fewer cases of postpartum sepsis, and more spontaneous vaginal deliveries occurred after partograph implementation [1]. In the Labour Progression Study, reported on in four articles [115–117,124], Zhang's labor curve and the WHO partograph were compared, revealing a difference in how oxytocin was used for augmentation, but not in the incidence of intrapartum CS, as well as longer labor durations with the use of Zhang's curve. In a study from Australia published after the review, the WHO's action line was compared with Neal and Lowe's stepped dystocia line [120]; the only difference in labor outcomes observed was a reduction in ARM in the stepped dystocia line group. In the final study not included in the Cochrane review, the effectiveness of 2- and 4-hour action lines was assessed [122]. The findings were comparable to those of the review [15]; neither design proved to be superior in preventing prolonged labor or improving birth outcomes [122].

## Discussion

In this scoping review, we mapped and described existing labor curves and the evidence for their effectiveness and accuracy. The main findings were: (1) that the majority of large studies describing labor curves were based on the US CSL cohort; (2) that labor progression was

described in most studies using the curve endpoint of the first measurement of 10 cm cervical dilatation; (3) that most assessments of curve accuracy were performed in low- and lower middle–income countries and none considered Zhang's curve; (4) that some curves have distinct clinical purposes; and (5) that the usefulness of labor curves is still a matter of debate, as studies have failed to prove their accuracy or effectiveness. We found few studies giving attention to the outcomes from women's perspectives.

## Contemporary labor curves are based mainly on three large US cohorts

We found that 20 of 73 studies were based on three large US cohorts [14,29,30,34,41,42,45,52,53,55,58,61,64,66,78,79,83,84,91,92], and data from the CSL (2002–2008) cohort were included in ten of the thirteen largest studies (comprising >50,000 participants), including that of Zhang et al. [4,14,34,41,52,53,55,66,83,91]. The routine care provided may alter physiologic labor progression and is context specific; examples include oxytocin augmentation and ARM [126–128]. The heterogeneity of routine care complicates the derivation of labor curves, as noted by Zhang et al. [14], who emphasized that the curves need to be interpreted within the context of current obstetric practice. Considering the massive nature of these studies relative to others, the US context needs to be considered when interpreting the generalizability and performance of labor curves.

## Most labor curves use the first measure of 10 cm cervical dilatation as the endpoint

In 69 of 73 studies included in this review, the first measurement of 10 cm cervical dilatation was used as the labor curve endpoint [4,7,8,14,26–66,70,72–95]. This 10-cm measure has been regarded as the end of the first stage of active labor since Friedman's [37] work in 1955. It does not reflect the end of labor, merely a step toward it and a point at which other parameters gain increased attention. In addition, as for all cervical dilatation steps, the exact timepoint at which dilatation reaches 10 cm is not known. Thus, the frequency of vaginal examination may affect the perception of the duration of the first stage of labor. The LCG recommends routine vaginal examination every 4 hours during active labor [17], and a reduced frequency could result in the documentation of longer first-stage labor durations. The most solid endpoint for labor, unbiased by subjective measures or differences in practices and guidelines, is birth. The time of vaginal birth served as the labor endpoint in only four studies included in this review, published in 1970–1990 [67–69,71].

## Recent studies of labor curve accuracy were conducted in low- and lower middle–income countries

Nineteen of 26 studies in which labor curve accuracy was assessed were conducted in low- and lower middle–income countries [3,7,8,32,81,96,97,99,100,102–110,113], and the systematic review on this topic was conducted in such countries and South Africa, an upper middle–income country [2]. Accuracy was not demonstrated clearly in these studies; however, oxytocin administration could be considered a potential moderator of adverse perinatal outcomes and rapid labor in such populations. A systematic review from low and lower middle-income countries on the use of oxytocin, found an association between the use of oxytocin and adverse perinatal outcomes [129]. The accuracy of labor curves may be compromised by local obstetric practice and precipitate oxytocin administration; the systematic review revealed that only 10.5% of women who received oxytocin had crossed the action line [129]. In addition, oxytocin administration may not be followed by adequate fetal monitoring in settings with limited

resources and staff. Maaløe et al. [130] found that the median time from the last recording of the fetal heart rate to delivery was 120 minutes prior to the implementation of new locally tailored guidelines and 74 minutes thereafter in Zanzibar; these intervals are far from meeting international recommendations for fetal monitoring [131]. Such practices may affect perinatal outcomes and the accuracy of labor curves for the detection of those at risk of adverse perinatal outcomes.

## Labor curves are developed for different purposes

The curves described and assessed in the studies included in this review were developed for different purposes, some of which were clearly stated. The alert and action lines developed by Philpott and Castle [7,8] were intended to distinguish "abnormal" labor, with the authors citing obstructed labor and the lack of competent staff as rationales for their need. Its main arena for utilization was primary health care facilities, striving for timely transfer of women at risk of prolonged labor to facilities offering more advanced labor care. Zhang et al. [14,93] developed their labor curve to combat the increase in CS performance in the US. The multicenter RCT of Bernitz et al. [115] revealed no reduction in intrapartum CSs with its use relative to WHO partograph use. Rather, the RCT showed significant reductions in CS performance in both groups, raising the question of whether an increased focus on progression reduces this practice [22]. We identified no study in which the accuracy of Zhang et al.'s [14] labor curve was assessed. Bonet et al. [2] examined the accuracy of the WHO alert line specifically as a diagnostic test for the identification of adverse birth outcomes in a systematic review, and stated that they were not able to assess the usefulness of the alert line in optimizing referral due to the lack of relevant studies.

## Strengths and limitations

Our scoping review has both strengths and limitations. We published the study protocol before commencing the literature searches [23]. We conducted comprehensive literature searches, making it unlikely that we missed any significant publications. The searches were not restricted by language limitations; however, during the screening process, we had to exclude certain studies to ensure that we only included those that the study group was confident about understanding. Our searches were confined to scientific literature, and the inclusion of guidelines and textbooks would likely have provided even more extensive descriptions of labor curves.

## Implications for future research

The scoping review approach enables the identification of research gaps. We found that the accuracy of labor curves has been assessed mainly in contexts in which labor care is challenged by a lack of resources and in which maternal and perinatal morbidity and mortality rates are high. According to our findings, the accuracy of labor curves has not been examined in a high-income setting since Chen & Chu's [31] 1986 study, performed with 143 women. Contemporary studies of labor curve accuracy in high-income settings would provide data to complement the existing evidence. Another research gap concerns birthing women's experiences, which were considered in few studies of labor curve effectiveness [15,118,119,121,122] and constitute a highly relevant outcome for positive childbirth experiences.

A range of labor curves has been presented for different populations, but their effectiveness and accuracy have not been convincingly verified. If labor curves fail to identify increased risks of adverse outcomes (perinatal, maternal, or increased intervention rates), we may need to rethink how they are used or find more useful tools to support decision making related to the

progression of labor. The mean and median are often taken to reflect "normal," but they reflect only central tendencies, not data distributions. In addition to studies of the usefulness of the LCG, future research on labor care should amplify the distribution of data to encompass the individuality of childbirth.

## Supporting information

**S1 Table. Characteristics of included studies.**
(PDF)

**S1 File. Initial literature search in Medline.**
(PDF)

**S2 File. Literature search history.** Detailed second search.
(PDF)

**S3 File. Data charting form.**
(PDF)

**S4 File. PRISMA-ScR checklist.**
(PDF)

**S5 File. Study protocol.**
(PDF)

## Acknowledgments

We thank Toril M. Hestnes, senior librarian at the University of Oslo, Library of Medicine and Science, for her invaluable help with performing the systematic literature search.

## Author Contributions

**Conceptualization:** Johanne Mamohau Egenberg Huurnink, Ellen Blix, Elisabeth Hals, Anne Kaasen, Stine Bernitz, Tina Lavender, Mia Ahlberg, Pål Øian, Aase Irene Høifødt, Andrea Solnes Miltenburg, Aase Serine Devold Pay.

**Data curation:** Johanne Mamohau Egenberg Huurnink, Ellen Blix, Elisabeth Hals, Anne Kaasen, Aase Irene Høifødt, Andrea Solnes Miltenburg, Aase Serine Devold Pay.

**Formal analysis:** Johanne Mamohau Egenberg Huurnink, Ellen Blix, Aase Serine Devold Pay.

**Funding acquisition:** Aase Serine Devold Pay.

**Investigation:** Johanne Mamohau Egenberg Huurnink, Ellen Blix, Elisabeth Hals, Anne Kaasen, Aase Irene Høifødt, Andrea Solnes Miltenburg, Aase Serine Devold Pay.

**Methodology:** Johanne Mamohau Egenberg Huurnink, Ellen Blix, Elisabeth Hals, Anne Kaasen, Stine Bernitz, Tina Lavender, Mia Ahlberg, Pål Øian, Aase Irene Høifødt, Andrea Solnes Miltenburg, Aase Serine Devold Pay.

**Project administration:** Johanne Mamohau Egenberg Huurnink, Aase Serine Devold Pay.

**Supervision:** Ellen Blix, Aase Serine Devold Pay.

**Visualization:** Johanne Mamohau Egenberg Huurnink.

**Writing – original draft:** Johanne Mamohau Egenberg Huurnink.

**Writing – review & editing:** Johanne Mamohau Egenberg Huurnink, Ellen Blix, Elisabeth Hals, Anne Kaasen, Stine Bernitz, Tina Lavender, Mia Ahlberg, Pål Øian, Aase Irene Høifødt, Andrea Solnes Miltenburg, Aase Serine Devold Pay.

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
