## [Decision Letter · Decision Letter 0]

7 Dec 2023

PONE-D-23-33373Labor curves based on cervical dilatation over time and their accuracy and effectiveness: a systematic scoping reviewPLOS ONE

Dear Dr. Huurnink,

Thank you for submitting your manuscript to PLOS ONE. After careful consideration, we feel that it has merit but does not fully meet PLOS ONE’s publication criteria as it currently stands. Therefore, we invite you to submit a revised version of the manuscript that addresses the points raised during the review process.

We look forward to receiving your revised manuscript.

Kind regards,

Thales Philipe Rodrigues da Silva, Ph.D

Academic Editor

PLOS ONE

Journal Requirements:

Additional Editor Comments:

Dear Huurnink.

Thank you for the opportunity to evaluate the article in question.

Thank you in advance.

In view of the reviewers' assessments, I am sending the article for further revision by the authors and future resubmission.

Sincerely,

Reviewers' comments:

Reviewer's Responses to Questions

**Comments to the Author**

1. Is the manuscript technically sound, and do the data support the conclusions?

Reviewer #1: Yes

Reviewer #2: Yes

2. Has the statistical analysis been performed appropriately and rigorously? 

Reviewer #1: N/A

Reviewer #2: Yes

3. Have the authors made all data underlying the findings in their manuscript fully available?

Reviewer #1: Yes

Reviewer #2: Yes

4. Is the manuscript presented in an intelligible fashion and written in standard English?

Reviewer #1: Yes

Reviewer #2: Yes

5. Review Comments to the Author

Reviewer #1: Thank you for an opportunity to review your paper. I read your paper with great interest and I see potential in your work that addresses current and relevant topics for maternal and child health. I have made some recommendations to strengthen your scoping review as follow:

1. Explicitly the exclusion criteria in the summary;

2. specify the methods used to assess the risk of bias;

3. standardize the study objectives in the summary and in the introduction of your paper;

4. present risk of bias assessments for each included study;

5. explain in detail methods for assessing confidence in the body of evidence for the outcomes analyzed.

Reviewer #2: Dear Authors and Editors: Thanks for the invitation to revise this manuscript. The objectives of this systematic scoping review were 1) map the existing evidence on labor curves that describe cervical dilatation over time; 2) map the studies providing evidence for and outcomes used to measure the curves’ effectiveness and accuracy; and 3) to identify research gaps. This is an interesting and well-written study with important conclusions and I recommend its publication in PLOS One.

6. PLOS authors have the option to publish the peer review history of their article (what does this mean?). If published, this will include your full peer review and any attached files.

Reviewer #1: No

Reviewer #2: No

---

## [Author Response · Author response to Decision Letter 0]

22 Dec 2023

Dear Editor and Reviewers,

Thank you for providing us with feedback and comments on the manuscript "Labor curves based on cervical dilatation over time and their accuracy and effectiveness: a systematic scoping review". We highly appreciate your contributions; it has helped us enhance our manuscript. Below, you will find our concise point-by-point response to the recommendations provided by reviewer #1. We sincerely hope that you will find the revised version suitable for publication.

Review #1:

1. Explicitly the exclusion criteria in the summary

a. We understand this as “explicitly state the exclusion criteria in the summary”. Exclusion criteria have been added to the abstract.

2. Specify the methods used to assess the risk of bias

a. According to Peters et al. [1], the scoping review methodology does not require assessment of risk of bias. This is due to the aim of scoping reviews of providing an overview of the field, regardless of study quality and potential bias. It can be relevant to conduct risk of bias assessment if a meta-analysis is conducted as part of a scoping review, however this is not typical. We did not conduct a meta-analysis as part of our scoping review and have therefore not assessed the risk of bias in the included studies.

3. Standardize the study objectives in the summary and in the introduction of your paper

a. The study objectives have been standardized accordingly in the summary and in the introduction.

4. Present risk of bias assessments for each included study

a. Please see response to question 2.

5. Explain in detail methods for assessing confidence in the body of evidence for the outcomes analyzed.

a. Methods for assessing confidence in the body of evidence for the outcomes analyzed is understood as linked to critical appraisal of the evidence. According to Peters et al. [1], critical appraisal of the evidence is not applicable for a scoping review. The confidence in the body of evidence in this scoping review, lies in the transparency and the systematic and comprehensive conduct and report of the review.

Due to the inherent nature of a scoping review, implications for practice are rarely made [1]. However, a scoping review can serve as a valuable tool to provide a comprehensive overview of the existing literature, highlighting what has been studied and published within the field of labor curves and identifying areas that require further investigation. The scoping review also includes systematic reviews where the included studies are assessed according to risk of bias. Our review can be seen as a complement to these meta-analyses, and provides a broad picture of the existing evidence, independent of heterogeneity and quality of studies.

Thank you once again for your valuable feedback. We sincerely hope that our revised manuscript complemented by our response to your suggestions, meet your expectations. We look forward to hearing from you soon.

Johanne M.E. Huurnink

References

1. Peters M, Godfrey C, McInerney P, Munn Z, Tricco A, Khalil H. Chapter 11: Scoping Reviews (2020 version). 2020. Joanna Briggs Institute Reviewer's Manual, JBI, Available from https://reviewersmanual.joannabriggs.org/. Available from: https://reviewersmanual.joannabriggs.org/

---

## [Decision Letter · Decision Letter 1]

17 Jan 2024

Labor curves based on cervical dilatation over time and their accuracy and effectiveness: a systematic scoping review

PONE-D-23-33373R1

Dear Dr. Huurnink,

We’re pleased to inform you that your manuscript has been judged scientifically suitable for publication and will be formally accepted for publication once it meets all outstanding technical requirements.

Kind regards,

Thales Philipe Rodrigues da Silva, Ph.D

Academic Editor

PLOS ONE

Additional Editor Comments:

Dear Authors,

I am pleased to inform you that manuscript has been accepted for publication. Congratulations!  

My comments, and any additional reviewer comments, can be found below.

Thanks to the authors for addressing all of the reviewer comments, I have accepted the paper for publication.

The only modification required will be to improve the quality of the image resolution.

Kind regards

Reviewers' comments:

Reviewer's Responses to Questions

**Comments to the Author**

1. If the authors have adequately addressed your comments raised in a previous round of review and you feel that this manuscript is now acceptable for publication, you may indicate that here to bypass the “Comments to the Author” section, enter your conflict of interest statement in the “Confidential to Editor” section, and submit your "Accept" recommendation.

Reviewer #1: All comments have been addressed

Reviewer #2: All comments have been addressed

2. Is the manuscript technically sound, and do the data support the conclusions?

Reviewer #1: Yes

Reviewer #2: Yes

3. Has the statistical analysis been performed appropriately and rigorously? 

Reviewer #1: N/A

Reviewer #2: Yes

4. Have the authors made all data underlying the findings in their manuscript fully available?

Reviewer #1: Yes

Reviewer #2: Yes

5. Is the manuscript presented in an intelligible fashion and written in standard English?

Reviewer #1: Yes

Reviewer #2: Yes

6. Review Comments to the Author

Reviewer #1: (No Response)

Reviewer #2: The manuscript is well-written and sound and all reviewers's comments and suggestions have been addressed. I recommend its approval and publication in PLOS One.

7. PLOS authors have the option to publish the peer review history of their article (what does this mean?). If published, this will include your full peer review and any attached files.

Reviewer #1: No

Reviewer #2: No

---

## [Editor Report · Acceptance letter]

14 Mar 2024

PONE-D-23-33373R1 

PLOS ONE

Dear Dr. Huurnink, 

I'm pleased to inform you that your manuscript has been deemed suitable for publication in PLOS ONE. Congratulations! Your manuscript is now being handed over to our production team.

Kind regards, 

on behalf of

Dr. Thales Philipe Rodrigues da Silva 

Academic Editor

PLOS ONE